# Transient non-integrative expression of nuclear reprogramming factors promotes multifaceted amelioration of aging in human cells

Tapash Jay Sarkar[1,2,3], Marco Quarta[4,5,6,7✉], Shravani Mukherjee[8], Alex Colville[4,5,6,9], Patrick Paine[4,5,6,7], Linda Doan[4,5,6,7], Christopher M. Tran[4,5,6], Constance R. Chu[8,10], Steve Horvath [11,12], Lei S. Qi [13], Nidhi Bhutani[8], Thomas A. Rando[4,5,6] & Vittorio Sebastiano[1,2✉]

Aging is characterized by a gradual loss of function occurring at the molecular, cellular, tissue and organismal levels. At the chromatin level, aging associates with progressive accumulation of epigenetic errors that eventually lead to aberrant gene regulation, stem cell exhaustion, senescence, and deregulated cell/tissue homeostasis. Nuclear reprogramming to pluripotency can revert both the age and the identity of any cell to that of an embryonic cell. Recent evidence shows that transient reprogramming can ameliorate age-associated hallmarks and extend lifespan in progeroid mice. However, it is unknown how this form of rejuvenation would apply to naturally aged human cells. Here we show that transient expression of nuclear reprogramming factors, mediated by expression of mRNAs, promotes a rapid and broad amelioration of cellular aging, including resetting of epigenetic clock, reduction of the inflammatory profile in chondrocytes, and restoration of youthful regenerative response to aged, human muscle stem cells, in each case without abolishing cellular identity.

[1] Institute for Stem Cell Biology and Regenerative Medicine, Stanford University School of Medicine, Stanford, CA 94305, USA. [2] Department of Obstetrics and Gynecology, Stanford University School of Medicine, Stanford, CA 94305, USA. [3] Department of Applied Physics, Stanford University School of Humanities and Sciences, Stanford, CA 94305, USA. [4] Department of Neurology and Neurological Sciences, Stanford University School of Medicine, Stanford, CA 94305, USA. [5] Paul F. Glenn Center for the Biology of Aging, Stanford University School of Medicine, Stanford, CA 94305, USA. [6] Center for Tissue Regeneration, Repair and Restoration, Veterans Affairs Palo Alto Health Care System, Palo Alto, CA 94304, USA. [7] Molecular Medicine Research Institute, Sunnyvale, CA 94085, USA. [8] Department of Orthopedic Surgery, Stanford University School of Medicine, Sanford, CA 94305, USA. [9] Department of Genetics, Stanford University School of Medicine, Stanford, CA 94305, USA. [10] VA Palo Alto Health Care System, Palo Alto, CA 94304, USA. [11] Department of Human Genetics David Geffen School of Medicine, University of California, Los Angeles, CA 90095, USA. [12] Department of Biostatistics, Fielding School of Public Health, UCLA, Los Angeles, CA 90095, USA. [13] Department of Bioengineering, Department of Chemical and Systems Biology, ChEM-H, Stanford University, Stanford, CA 94305, USA. ✉email: mquarta@stanford.edu; vsebast@stanford.edu

The process of nuclear reprogramming to induced pluripotent stem cells (iPSCs) is characterized, upon completion, by a profound resetting of the epigenetic landscape of cells of origin, resulting in reversion of both cellular identity and age to an embryonic-like state[1–4].

Notably, if the expression of the reprogramming factors is only transiently applied and then stopped (before the so-called Point of No Return, PNR)[5], the cells return to the initiating somatic cell state. These observations suggest that, if applied for a short enough time, the expression of reprogramming factors fails to erase the epigenetic signature defining cell identity; however, it remains unknown whether any substantial and measurable reprogramming of cellular age can be achieved before the PNR. First evidence that transient reprogramming can promote amelioration of aging phenotypes was shown by Ocampo et al., in progeroid mice carrying a Dox-inducible OSKM cassette[6]. Yet, important questions remain open. Murine genetic models of premature aging only in part recapitulate the complexity of natural aging, a phenomenon that is characterized by a slow and progressive accumulation of epigenetic errors. In addition, proof is lacking that the same rejuvenative effect can be achieved with naturally aged human cells isolated from elderly individuals, together with a comprehensive molecular and physiological analysis of the depth and extension of the rejuvenation in human cells. To address all these questions, we devised a platform that could let us test whether transient expression of nuclear reprogramming genes has any impact in ameliorating aging phenotypes in naturally aged human and mouse cells across multiple cell types and spanning all the hallmarks of aging.

## Results

We first evaluated the effect of transient expression of reprogramming factors on the transcriptome of two distinct cell types—fibroblasts and endothelial cells—from aged human subjects, and we compared it with the transcriptome of the same cell types isolated from young donors (Fig. 1a, e). Fibroblasts were derived from arm and abdomen skin biopsies (25–35 years for the young control, $n = 3$, and 60–90 years for the aged group, $n = 8$), while endothelial cells were extracted from iliac vein and artery (15–25 years for the young control, $n = 3$, and 50–65 years for the aged group, $n = 7$). We utilized a non-integrative reprogramming protocol that we optimized, based on a cocktail of mRNAs expressing OCT4, SOX2, KLF4, c-MYC, LIN28, and NANOG (OSKMLN)[7]. Our protocol consistently produces iPSC colonies, regardless of age of the donors, after 12–15 daily transfections; we reasoned that the PNR in our platform occurs at about day 5 of reprogramming, based on the observation that the first detectable expression of endogenous pluripotency-associated lncRNAs occurs at day 5[8]. Therefore, we adopted a transient exogenous expression regimen where OSKMLN was daily transfected for 4 consecutive days, and performed gene expression analysis 2 days after the interruption (Fig. 1b).

We performed paired-end bulk RNA sequencing on both cell types for the same three cohorts: young (Y), untreated aged (UA), and treated aged (TA). First, we compared the quantile normalized transcriptomes of young and untreated aged cells for each cell type (Y vs. UA) and found that 961 genes (5.85%) in fibroblasts (678 upregulated, 289 downregulated, Fig. 1a, c) and 748 genes (4.80%) in endothelial cells (389 upregulated, 377 downregulated, Fig. 1e, f) differed between young and aged cells, with the significance criteria of $p < 0.05$ and a log fold change cutoff ±0.5 (full list of genes in Supplementary Data 1 and 2). We found these sets of genes were enriched for many of the known aging pathways, identified in the hallmark gene set collection in the Molecular Signatures Database[9] (Supplementary Data 3 and 4).

When we mapped the directionality of expression above or below the mean of each gene, we could observe a clear similarity between treated and young cells as opposed to aged cells for both fibroblasts and endothelial cells (Fig. 1d, g). We further performed principal component analysis in this gene set space and determined that the young and aged populations were separable along the first principal component (PC1), which explained 64.8% of variance in fibroblasts and 60.9% of variance in endothelial cells. Intriguingly, the treated cells also clustered closer to the younger cells than the aged cells, simply along PC1 (Supplementary Fig. 1a, b).

Using the same significance criteria defined above, we then compared the treated and untreated aged populations (TA vs. UA) (Fig 1a, e, Supplementary Fig. 2 and Supplementary Data 5 and 6) and found that 1042 genes in fibroblasts (734 upregulated and 308 downregulated) and 992 in endothelial cells (461 upregulated and 531 downregulated) were differentially expressed. Interestingly, also within these sets of genes, we found enrichment for aging pathways, within the Molecular Signatures Database[9] as previously described (Supplementary Data 7 and 8). When we compared the profiles young versus untreated aged (Y vs. UA) and untreated aged versus treated aged (UA vs. TA) in each cell type, we observed a 24.7% overlap for fibroblasts (odds ratio of 4.53, $p < 0.05$) and 16.7% overlap for endothelial cells (odds ratio of 3.84, $p < 0.05$) with the directionality of change in gene expression matching that of youth (i.e., if higher in young then higher in treated aged); less than 0.5% moved oppositely in either cell types (Supplementary Fig. 1a, b and Supplementary Data 9 and 10).

Next, we used these transcriptomic profiles to verify retention of cell identity after treatment. To this end, using established cell identity markers, we verified that none significantly changed upon treatment (Supplementary Data 11). In addition, we could not detect the expression of any pluripotency-associated markers (other than the OSKMLN mRNAs transfected in) (Supplementary Data 11). Altogether, the analysis of the transcriptomic signatures revealed that OSKLMN expression promotes a very rapid activation of a more youthful gene expression profile, which is cell-type specific, without affecting the expression of cell identity genes.

Epigenetic clocks based on DNA methylation levels are the most accurate molecular biomarkers of age across tissues and cell types and are predictive of a host of age-related conditions including lifespan[3,10–12]. Exogenous expression of canonical reprogramming factors (OSKM) is known to revert the epigenetic age of primary cells to a prenatal state[3]. To test whether transient expression of OSKMLN could reverse the epigenetic clock of human somatic cells, we used two epigenetic clocks that apply to human fibroblasts and endothelial cells: Horvath's original pan-tissue epigenetic clock (based on 353 cytosine–phosphate–guanine pairs), and the more recent skin-and-blood clock (based on 391 CpGs)[3,13].

According to the pan-tissue epigenetic clock, transient OSKMLN significantly (two-sided mixed-effect model $P$ value = 0.023) reverted the DNA methylation age (average age difference = −3.40 years, standard error 1.17). The rejuvenation effect was more pronounced in endothelial cells (average age difference = −4.94 years, SE = 1.63, Fig. 1i) than in fibroblasts (average age difference = −1.84, SE = 1.46, Fig. 1h). Qualitatively similar, but less significant results could be obtained with the skin-and-blood epigenetic clock (overall rejuvenation effect −1.35 years, SE = 0.67, one-sided mixed-effect model $P$ value = 0.042, and average rejuvenation in endothelial cells and fibroblasts is −1.62 years and −1.07, respectively).

Prompted by these results, we next analyzed the effect of transient reprogramming on various hallmarks of cellular

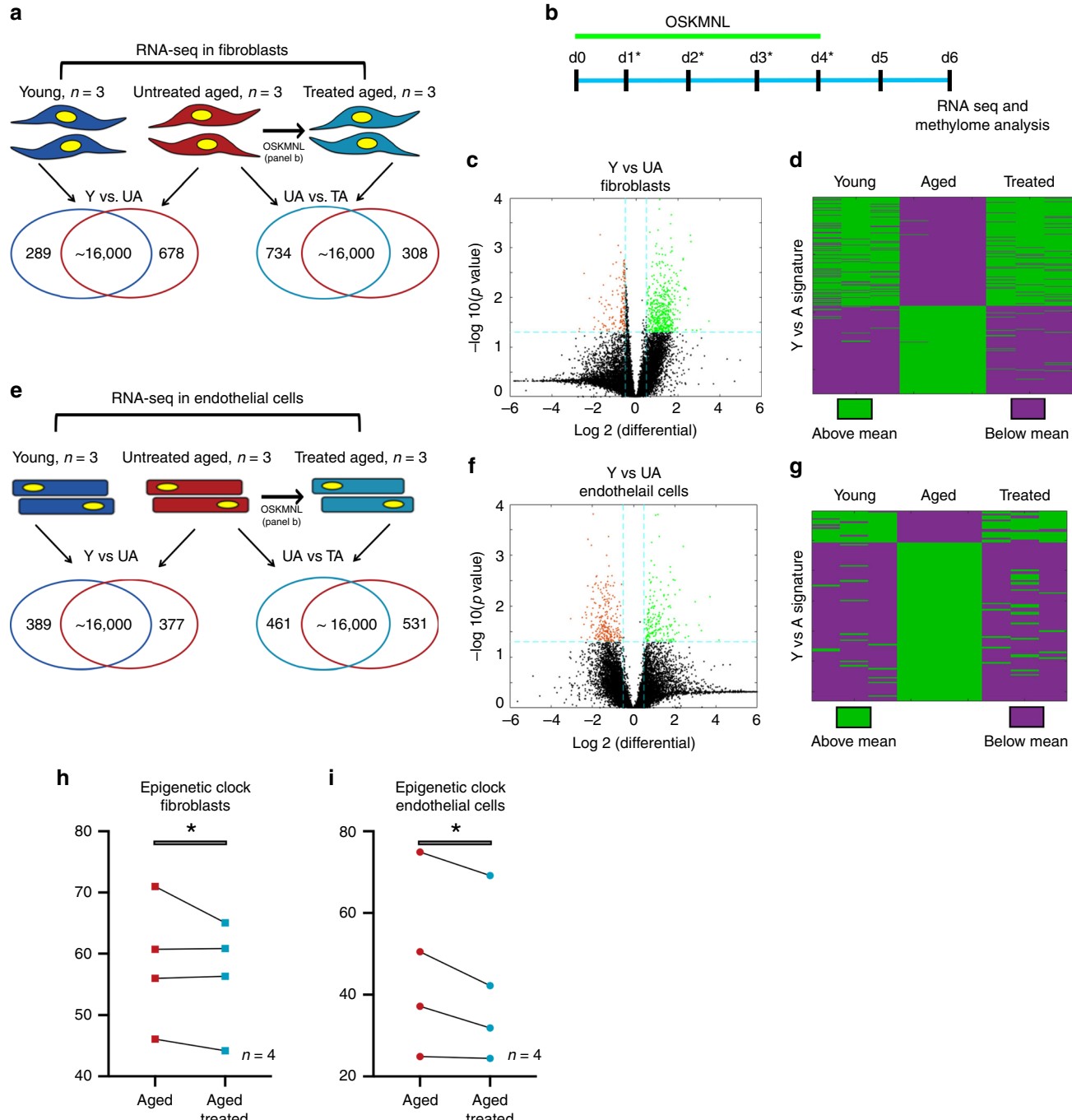

**Fig. 1 Transcriptomic and epigenetic clock analysis shows more youthful signature upon transient expression of OSKMNL in human fibroblasts and endothelial cells. a** Venn diagrams show differentially expressed genes in fibroblasts (young, $n = 3$ individuals; aged and aged treated $n = 3$ individuals) defined with at significance $p$ value >0.05 and log fold change >0.5. Comparison among the three groups was conducted by ANOVA test. **b** Schematic of reprogramming protocol. **c** Volcano plot showing young versus aged fibroblast differential gene expression. **d** Heat map of polarity of expression (green = above, purple = below) the mean for each differential gene. The distribution shows the treated samples transition in expression in this space towards the direction of the young fibroblasts. Cells in each cohort were subjected to 80 bp paired-end reads of RNA sequencing and quantile normalized. **e** Venn diagrams show differentially expressed genes in endothelial cells (young, $n = 3$ individuals; aged and age-treated $n = 3$ individuals) defined at significance $p$ value >0.05 and log fold change >0.5. Comparison among the three groups was conducted by ANOVA test. **f** Volcano plot showing young versus aged endothelial cells differential gene expression. **g** Heat map of polarity of expression (green = above, purple = below) the mean for each differential gene. The distribution shows the treated samples transition in expression in this space towards the direction of the young endothelial cells. **h** Methylation clock estimation of patient sample age with and without treatment for fibroblasts; $n = 4$ individuals. **i** Methylation clock estimation of patient sample age with and without treatment for endothelial cells; $n = 4$ individuals. Statistical analysis of methylation clock was performed by two-sided $t$-test analysis.

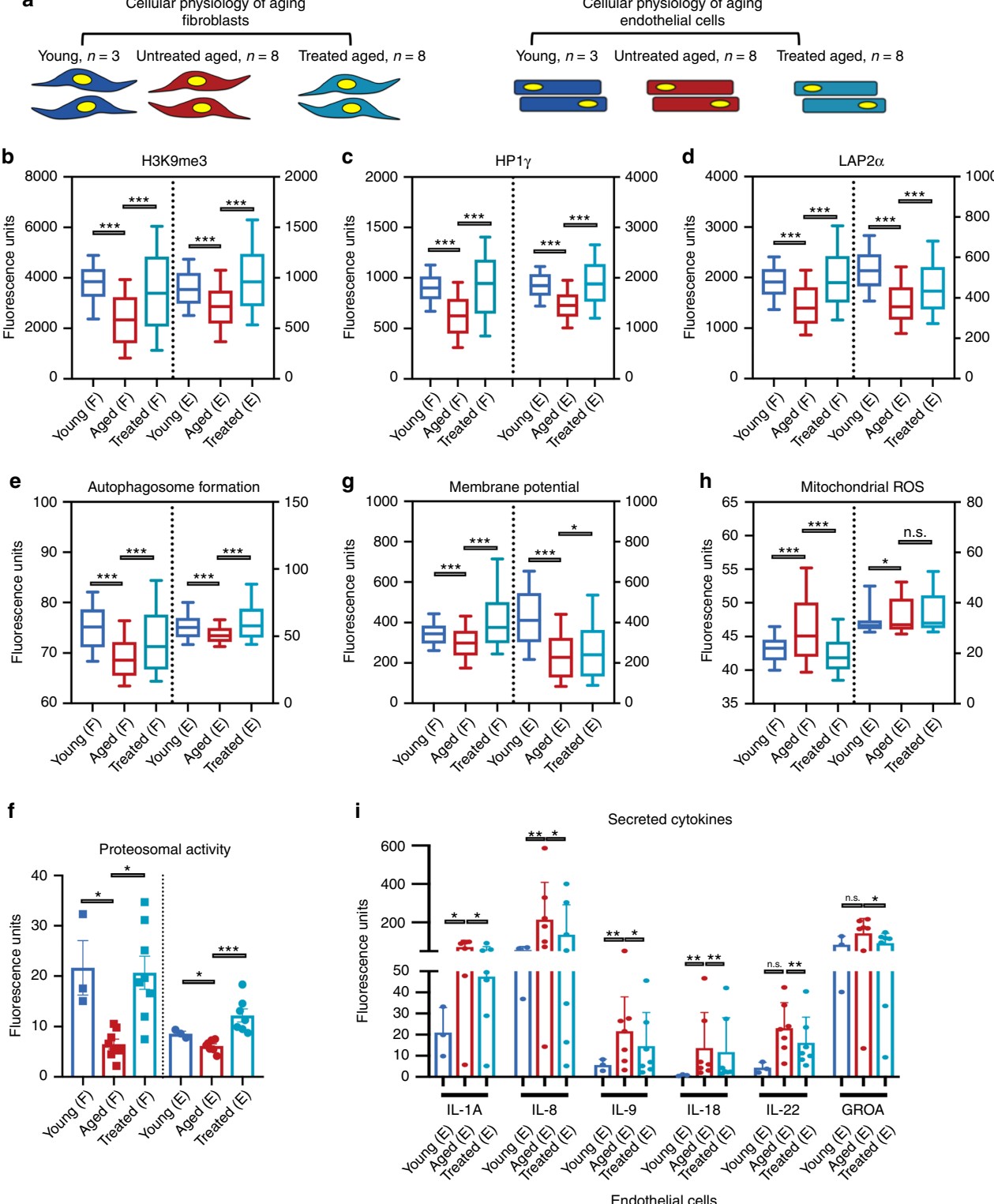

physiological aging. We employed a panel of 11 established assays, spanning the hallmarks of aging[14] (Supplementary Data 12), and performed most of the analyses using single-cell high-throughput imaging to capture quantitative changes in single cells and distribution shifts in the entire population of cells. All the analyses were performed separately in each individual cell line (total of 19 fibroblast lines: 3 young, 8 aged, and 8 treated aged; total 17 endothelial cell lines: 3 young, 7 aged, and 7 treated aged) (Fig. 2a and Supplementary Figs. 2–5). Statistical analysis

was conducted by randomly sampling 100 cells per sample; the data was subsequently pooled by group category (see "Materials and Methods" for a detailed description of the Statistical methods that were used). Control experiments were performed by adopting the same transfection scheme using mRNA encoding for green fluorescent protein (GFP) (Supplementary Figs. 6 and 7).

To extend our previous findings on epigenetics, we quantitatively measured by immunofluorescence (IF) the epigenetic repressive mark H3K9me3, the heterochromatin-associated

**Fig. 2 Transient OSKMNL expression reverts aged physiology toward a more youthful state in human fibroblasts and endothelial cells. a** Fibroblasts (F) and endothelial cells were obtained from otherwise healthy young and aged individuals. Young untreated cells ($n = 3$ distinct individuals for both fibroblasts and endothelial cells, dark blue), aged untreated cells ($n = 8$ individuals for fibroblast, $n = 7$ individuals for endothelial cells, red), and aged treated cells ($n = 8$ for fibroblast, $n = 7$ for endothelial cells, light blue) were analyzed for a panel of 11 different hallmarks of aging. Most of the assays were performed by high-throughput imaging on 500–1000 cells per sample to allow population-wide studies with single-cell resolution (Supplementary Figs. 2–5). 100 cells per sample (i.e., individuals) were randomly selected and pooled per treatment group to do a statistical comparison across the three groups (young fibroblasts $n = 300$; aged fibroblasts $n = 800$; aged treated fibroblasts $n = 800$; young endothelial cells $n = 300$; aged endothelial cells $n = 700$; aged treated endothelial cells $n = 700$). Pairwise statistical analysis was done by one-way ANOVA. *$P < 0.05$, **$P < 0.01$, ***$P < 0.001$. **b** Quantification of single-nucleus levels of trimethylated H3K9, a repressive mark of gene expression. Both cell types show significant elevation of the mark towards the youthful distribution. **c** Quantification of single-nucleus levels of heterochromatin marker HP1γ by immunocytochemistry showing a trend toward youth upon treatment. **d** Quantification of the inner nuclear membrane polypeptide LAP2α, a regulator of nuclear lamina by regulating the binding of lamin B1 and chromatin. This again shows a trend toward youth after cells are treated. **e** Results of live cells imaging with florescent marker of autophagosome formation in single cells. **f** Cleavage of fluorescent-tagged chymotrypsin-like substrate elevated in treated and young fibroblasts and endothelial cells corresponding to increased proteasome 20S core particle activity. **g** Individual cell mitochondria membrane potential measurements also showing more active mitochondria as a result of transient reprogramming. Quantification of pro-inflammatory factors secreted by the cells in each cohort. **h** Individual cell mitochondria ROS measurements also showing less accumulated ROS as a result of transient reprogramming. **i** Inflammatory cytokine profiling in endothelial cells, with a significant elevation and depression specifically in aged and treated endothelial cells, respectively. In **b–h** data are represented as box–whisker plots with median, and bars represent whiskers with distribution variability 10th–90th percentile. In **f–j** data are represented as mean values and bars represent SD.

protein HP1γ, and the nuclear lamina support protein LAP2α (Fig 2b–d). Aged fibroblasts and endothelial cells showed a decrease in the nuclear signal for all three markers compared with young cells, as previously reported[2]; treatment of aged cells resulted in an increase of these markers in both cell types. Next, we examined both pathways involved in proteolytic activity of the cells by measuring formation of autophagosomes, and chymotrypsin-like proteasomal activity, both reported to decrease with age[15,16]. Treatment increased both to levels similar to or even higher than young cells, suggesting that early steps in reprogramming promote an active clearance of degraded biomolecules (Fig. 2e, f).

In terms of energy metabolism, aged cells display decreased mitochondrial activity, accumulation of reactive oxygen species (ROS), and deregulated nutrient sensing[2,16,17]. We therefore tested the effects of treatment on aged cells by measuring mitochondria membrane potential, mitochondrial ROS, and levels of Sirtuin1 protein (SIRT1) in the cells. Transient reprogramming increased mitochondria membrane potential in both cell types (Fig. 2g), while it decreased mitochondrial ROS (Fig. 2h) and increased SIRT1 protein levels in fibroblasts, similar to young cells (Supplementary Fig. 8). Senescence-associated beta-galactosidase staining showed a significant reduction in the number of senescent cells in aged endothelial cells but not in fibroblasts (Supplementary Fig. 8). This decrease was accompanied by a decrease in pro-inflammatory senescence-associated secretory phenotype cytokines again in endothelial cells and not in fibroblasts (Fig. 2i and Supplementary Fig. 8)[16,18,19]. Last, in neither cell type did telomere length, measured by quantitative fluorescence in situ hybridization[2,20], show significant extension with treatment (Supplementary Fig. 8), suggesting that the cells did not dedifferentiate into a stem-like state in which telomerase activity would be reactivated, and in agreement with previous reports where activation of TERT was observed at later stages of nuclear reprogramming[21].

Next, we assessed the perdurance of these effects and found that most were significantly retained after 4 and 6 days from the interruption of reprogramming (Supplementary Figs. 9 and 10). We then examined how rapidly these physiological rejuvenative changes manifest by repeating the same sets of experiments in fibroblasts and endothelial cells that were transfected for just 2 consecutive days. Remarkably, we observed that most of the rejuvenative effects could already be seen after 2 days of treatment, although most were more moderate (Supplementary Figs. 11 and 12).

Collectively, this data demonstrates that transient expression of OSKMLN can induce a rapid, persistent amelioration, and reversal of cellular age in human somatic cells at the transcriptomic, epigenetic, and cellular levels. Importantly, these data demonstrate that the process of cellular rejuvenation is engaged very early, rapidly, and broadly in the reprogramming process. These epigenetic and transcriptional changes occur before any epigenetic reprogramming of cellular identity takes place, a novel finding in the field.

With these indications of a beneficial effect on cellular aging, we next investigated whether transient expression of OSKMNL could also reverse the inflammatory phenotypes associated with aging. After obtaining preliminary evidence of this reversal in endothelial cells (Fig. 2j), we extended our analysis to osteoarthritis, a disease strongly associated with aging and characterized by a pronounced inflammatory spectrum affecting the chondrocytes within the joint[22]. We thus isolated chondrocytes from cartilage of six 60–70-year-old patients undergoing total joint replacement surgery owing to their advanced-stage OA, and compared the results of treatment with chondrocytes isolated from three young individuals (Fig. 3a). Transient OSKMLN expression was performed for 2 or 3 days, and the analysis performed after 2 days from interruption of reprogramming, though the more consistent effect across patients was with longer treatment. Treatment showed a significant reduction in intracellular mRNA levels of RANKL and iNOS2, as well as in levels of inflammatory factors secreted by the cells (Fig. 3b–d). In addition, we observed increased cell proliferation (Fig. 3e), increased ATP production (Fig. 3f), and decreased oxidative stress as revealed by reduced mitochondrial ROS and elevated RNA levels of antioxidant SOD2 (Fig. 3g, h), a gene that has been shown to be downregulated in OA[23]. Finally, when we checked for retention of cellular identity, we observed that the treatment did not affect the expression level of SOX9 (a transcription factor core to chondrocyte identity and function) and significantly increased the level of expression of COL2A1 (the primary collagen in articular cartilage) (qRT-PCR in Fig 3i, j), suggesting retention of chondrogenic cell identity. Together, these results show that transient expression of OSKMLN can promote a partial reversal of gene expression and cellular physiology in aged OA chondrocytes toward a healthier, more youthful state, suggesting a potential new therapeutic strategy to ameliorate the OA disease process.

Stem cell loss of function and regenerative capacity represents another important hallmark of aging[14]. We thus wanted to assess

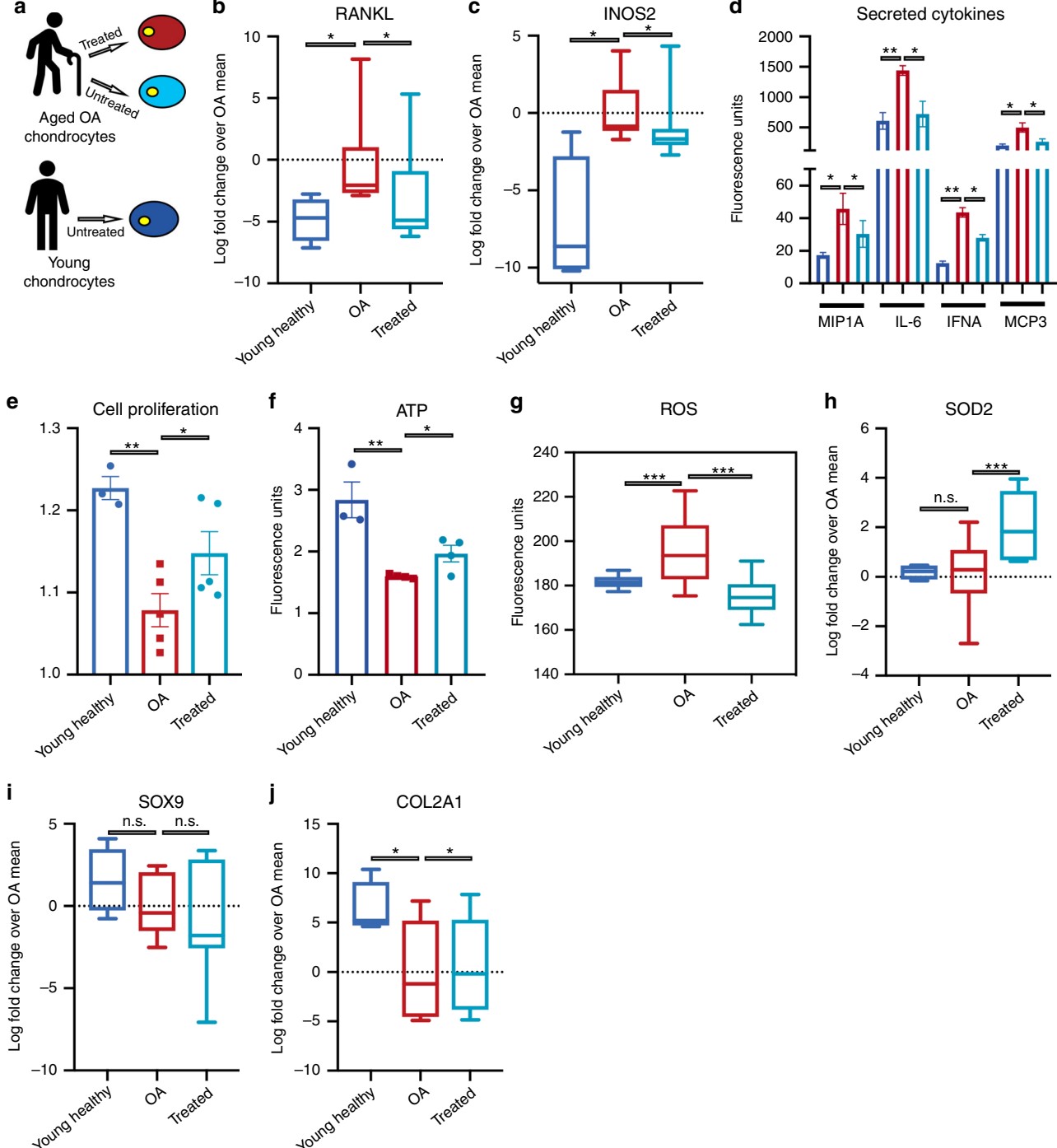

**Fig. 3 Transient OSKMNL expression mitigates inflammatory phenotypes in diseased chondrocytes. a** Workflow summarizing the strategy adopted to mitigation of age-related disease. Chondrocytes were obtained from six distinct aged patients diagnosed late stage Osteoarthritis (OA) patients from cartilage biopsies. Healthy cells (blue), aged OA cells (red) and transiently reprogrammed OA cells (light blue) were evaluated for OA specific phenotypes. **b** qRT-PCR evaluation shows treatment diminishes of intracellular RNA levels of NF-κB ligand RANKL. **c** qRT-PCR evaluation shows treatment drops levels of iNOS for producing nitric oxide as a response and to propagate inflammatory stimulus. **d** Cytokine profiling of chondrocyte secretions shows an increase pro-inflammatory cytokines in OA chondrocytes that diminishes with treatment. **e** Cell proliferation rate as measured by cell-tracking dye. **f** Measurement of ATP concentration using glycerol based fluorophore shows elevation of ATP levels with treatment. **g** Live single-cell image of cells up taking superoxide triggered fluorescent dyes shows diminished signal after treatment. **h** qRT-PCR evaluation of RNA levels of antioxidant SOD2, elevated with treatment. **i** qRT-PCR levels of chondrogenic identity and function transcription factor SOX9 is retained after treatment. **j** qRT-PCR shows elevation RNA levels for extracellular matrix protein component. Young samples $n = 3$ individuals; aged OA samples treated and untreated $n = 6$ individuals. Pairwise statistical analysis was done by one-way ANOVA. For ROS (**g**) analysis was conducted by high-throughput imaging on 500–1000 cells per sample to allow population-wide studies with single-cell resolution. One-hundred cells per sample were randomly selected to do a statistical comparison across the three groups. Statistical analysis was then done by one-way ANOVA. *$P < 0.05$, **$P < 0.01$, ***$P < 0.001$. Statistical analysis by one-way ANOVA was conducted for all the other assays.

the effect of transient reprogramming on the age-related changes in somatic stem cells that impair regeneration. First, we tested the effect of transient reprogramming on mouse-derived skeletal muscle stem cells (MuSCs). We treated MuSCs for 2 days while they were kept in a quiescent state using an artificial niche[24]. We conducted initial experiments with young (3 month) and aged (20–24 months) murine MuSCs isolated by FACS (Fig. 4a). Treatment of aged MuSCs reduced both time of first division, approaching the faster activation kinetics of quiescent young MuSCs[25,26], and mitochondrial mass[27] (Supplementary Fig. 13a, b). Moreover, treatment partially rescued the reduced ability of single MuSCs to form colonies[25,28] (Supplementary Fig. 13c). We further cultured these cells and observed that treatment did not change expression of the myogenic marker MyoD but instead improved their capacity to differentiate into myotubes (Supplementary Fig. 14a–d), suggesting that transient reprogramming does not disrupt the myogenic fate but can enhance the myogenic potential.

Next, we wanted to test MuSC function and potency to regenerate new tissue in vivo. To do this, we transduced young, aged, or transiently reprogrammed aged MuSCs with a lentivirus-expressing luciferase and GFP, and then transplanted the cells into injured tibialis anterior (TiA) muscles of immunocompromised mice. Longitudinal bioluminescence imaging (BLI) initially showed that muscles transplanted with treated aged MuSCs showed the highest signal (day 4, Fig 4b, c), but became comparable with muscles with young MuSCs by day 11 post transplantation; conversely muscle with untreated aged MuSCs showed lower signals at all time points post transplantation (Fig. 4b, c). IF analysis further revealed higher numbers of donor-derived (GFP+) myofibers in TiAs transplanted with treated compared with untreated aged MuSCs (Fig. 4d, e). Moreover, the GFP+ myofibers from treated aged cells exhibited increased cross-sectional areas when compared with their untreated counterparts, and in fact even larger than the young controls (Fig. 4f). Together, these results suggest improved tissue regenerative potential of transiently reprogrammed aged MuSCs. After 3 months, all mice were subjected to autopsy, and no neoplastic lesions or teratomas were discovered (Supplementary Table 1). To test potential long-term benefits of the treatment, we induced a second injury 60 days after cell transplantation, and again observed that TiA muscles transplanted with transiently reprogrammed aged MuSCs yielded higher BLI signals (Fig. 4g).

Sarcopenia is an age-related condition that is characterized by loss of muscle mass and force production[29,30]. Similarly, in mice muscle functions show progressive degeneration with age[31,32]. We wanted to test whether transient reprogramming of aged MuSCs would improve a cell-based treatment in restoring physiological functions of muscle of older mice. To test this, we first performed electrophysiology to measure tetanic force production in TiA muscles isolated from young (4 months) or aged (27 months) immunocompromised mice. We found that TiA muscles from aged mice have lower tetanic forces compared with young mice, suggesting an age-related loss of force production (Fig. 4h). Next, we isolated MuSCs from aged mice (20–24 months). After treating aged MuSCs, we transplanted them into cardiotoxin-injured TiA muscles of aged (20 months) immunocompromised mice. We waited 30 days to give enough time to the transplanted muscles to fully regenerate. We then performed electrophysiology to measure tetanic force production. Muscles transplanted with untreated aged MuSCs showed forces comparable with untransplanted muscles from aged control mice (Fig. 4h). Conversely, muscles that received treated aged MuSCs showed tetanic forces comparable with untransplanted muscles from young control mice (Fig. 4h and Supplementary Fig. 15a). These results suggest that transient reprogramming in combination with MuSC-based therapy can restore physiological function of aged muscles to that of youthful muscles.

Last, we wanted to translate these results to human MuSCs. We repeated the study, employing operative samples obtained from patients in different age ranges (10–80 years old), and transducing them with GFP- and luciferase-expressing lentiviral vectors (Fig. 4a). As in mice, transplanted, transiently reprogrammed, aged human MuSCs resulted in increased BLI signals compared with untreated MuSCs from the same individual, and comparable with those observed with young MuSCs (Fig. 4i and Supplementary Fig. 16a, b). Interestingly, the BLI signal ratio between contralateral muscles with treated and untreated MuSCs was higher in the older age group (60–80 years old) than in the younger age groups (10–30 or 30–55 years old), suggesting that ERA restores lost functions to younger levels in aged cells (Fig. 4j). Taken together, these results suggest that transient reprogramming partially restores the potency of aged MuSCs to a degree similar to that of young MuSCs, without compromising their fate, and thus has potential as a cell therapy in regenerative medicine.

Nuclear reprogramming to iPSCs is a multi-phased process comprising initiation, maturation, and stabilization[33]. Upon completion of such a dynamic and complex epigenetic reprogramming, iPSCs are not only pluripotent but also youthful. While proof of principle that transient reprogramming can exert a systemic rejuvenation in a genetic model of aging (progeroid mice), the proof that a multispectral cellular rejuvenation could be achieved in a cell-autonomous fashion in human cells isolated from naturally aged individuals was missing. Here we demonstrate that a non-integrative, mRNAs-based platform of transient cellular reprogramming can very rapidly reverse a broad spectrum of aging hallmarks in the initiation phase, when epigenetic erasure of cell identity has not yet occurred. We show that the process of rejuvenation occurs in naturally aged human and mouse cells, with restoration of lost functionality in diseased cells and aged stem cells while preserving cellular identity. Future studies are required to elucidate the mechanism that drives the reversal of the aged phenotype during cellular reprogramming, uncoupling it from dedifferentiation process[34,35]. Our results are novel and represent a significant step toward the goal of reversing cellular aging, and have potential therapeutic implications for aging and aging-related diseases.

## Methods

**Human fibroblast isolation and culture.** Isolation was performed at Coriell Institute on healthy patients and from Alzheimer patient samples at Stanford Hospital, in accordance to the methods and protocols approved by the Institutional Review Board of Stanford University, biopsied for skin mesial aspect of mid-upper arm or abdomen using 2-mm punch biopsies from both male and female patients 60–70 years old ($n = 8$) and 25–35 years ($n = 3$). Cells were cultured out from these explants and maintained in Eagle's Minimum Essential Medium with Earl's salts supplemented with nonessential amino acids, 10% fetal bovine serum, and 1% Penicillin/Streptomycin. Cells were cultured at 37 °C with 5% $CO_2$.

**Human endothelial cell isolation and culture.** Isolation was performed at Coriell Institute from iliac arteries and veins, and muscle biopsies from Stanford Hospital, in accordance to the methods and protocols approved by the Institutional Review Board of Stanford University, from otherwise healthy 45–60 years old ($n = 7$). Tissue was digested with collagenase and cells released from the lumen were used to initiate cultures. Plates for seeding were coated with 2% gelatin, then washed with PBS before use. Cells were maintained in Medium 199 supplemented with 2 mM L-glutamine, 15% fetal bovine serum, 0.02 mg/ml Endothelial Growth Supplement, 0.05 mg/ml Heparin, and 1% Penicillin/Streptomycin. Cells were cultured at 37 °C with 5% $CO_2$.

**Human articular chondrocyte isolation and culture.** In accordance to the methods and protocols approved by Institutional Review Board of Stanford University, the human OA chondrocytes were derived from discarded tissues of OA patients (50–72 years of age, $n = 6$) undergoing total knee arthroplasty. The samples were surgical waste and were fully deidentified prior to procurement,

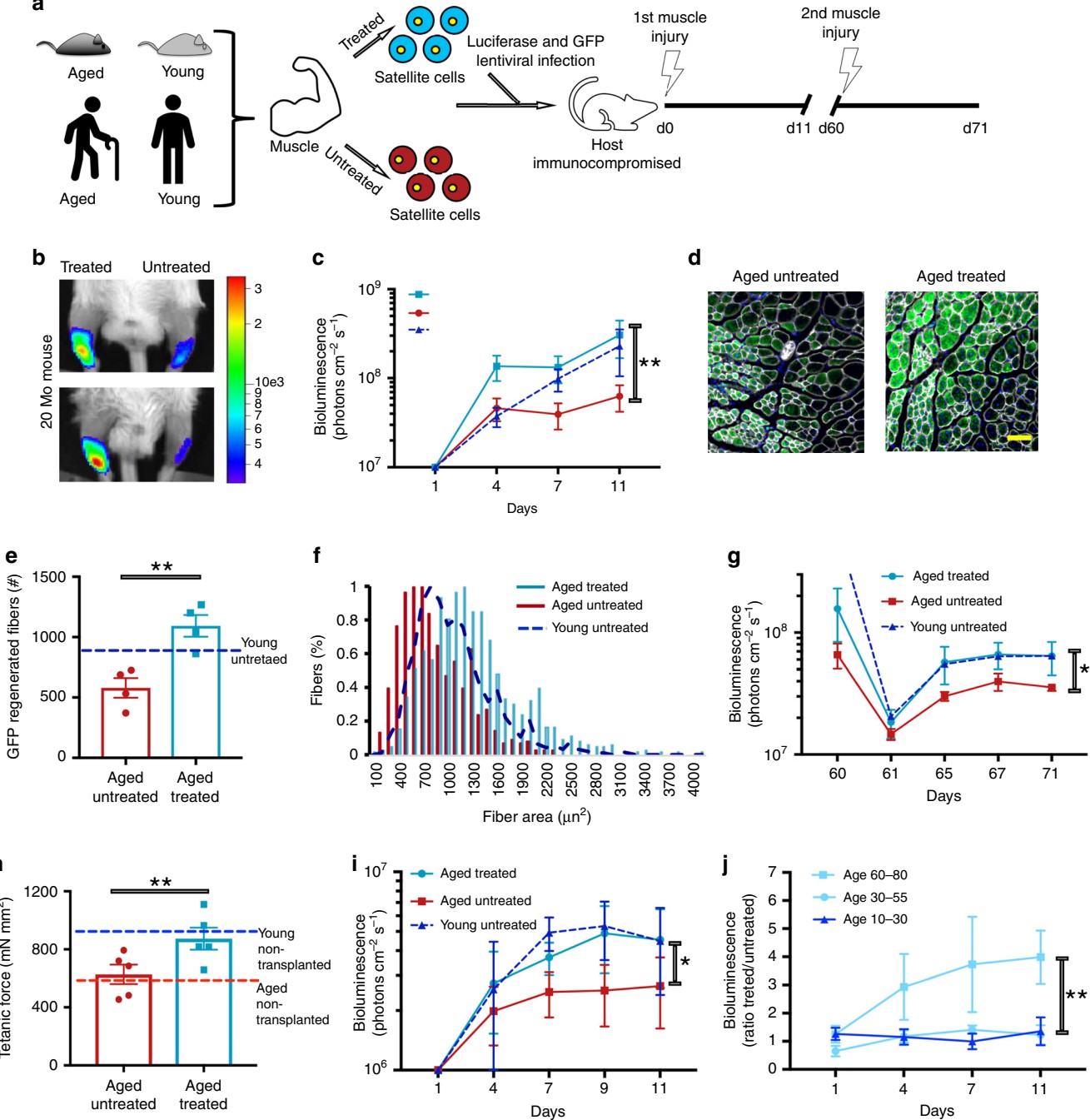

**Fig. 4 Transient OSKMNL expression restores aged muscle stem cell potency. a** Schematic showing the experimental design of partially reprogrammed aged mouse and human MuSCs. **b** Representative images of bioluminescence measured from mice 11 days after transplantation and injury in TiA muscles of treated/untreated Luciferase+ mouse MuSCs. **c** Quantified results of bioluminescence in **b** at different time points following transplantation and injury (*n* = 10). **d** Representative immunofluorescence of GFP expression in TiA muscle cross-sections of mice imaged and quantified in **c** and **d**, isolated 11 days after transplantation (Scale bar = 500 μm). **e** Quantification of immunofluorescence staining in **d** (*n* = 5). **f** Quantification of the cross-sectional area of donor-derived GFP + fibers in TiA muscles that were recipients of transplanted MuSCs (*n* = 5). **g** Results of bioluminescence imaging of TiA muscles reinjured after 60 days (second injury) after MuSC transplantations (*n* = 6). The second injury was performed to test whether the bioluminescence signal increased as a consequence of activating and expanding luciferase+/GFP+ MuSCs that were initially transplanted and that had engrafted under the basal lamina. **h** Tetanic force measurements of aged muscles injured and transplanted with aged MuSCs. TiA muscles were dissected and electrophysiology ex vivo for tetanic measurement performed. Baseline of force production of untransplanted muscles was measured in young (4 months, blue broken line) and aged (27 months, red broken line) mice. Treated aged MuSCs were transplanted into TiA muscles of aged mice and force production measured 30 days later (*n* = 5). **i** Quantified results of bioluminescence measured from mice 11 days after transplantation in TiA muscles of treated Luciferase+ human MuSCs. **j** Variation in ratio of bioluminescence between treated and untreated MuSCs obtained from healthy donors of different age groups. Significance is calculated with one-sided student's *t* test, pairwise between treated and aged, and group wise when comparing with young patients (age groups. 10–30: *n* = 5 individuals; 30–55: *n* = 7 individuals; 60–80: *n* = 5 individuals). *P < 0.05, **P < 0.01, ***P < 0.001, color of the asterisks matches the population being compared with.

hence no prior patient consent was required. Cartilage pieces were shaved off bone by scalpel, taking care to avoid any fat, then digested with collagenase in DMEM/F12 media (supplemented with 25 mg/ml ascorbate, 2 mM L-glutamine, 1% penicillin/streptomycin antibiotics, and 10% fetal bovine serum) for 1–2 days until shavings were substantially dissolved. Supernatant from cultures was strained, filtered, and centrifuged, and the cells were then resuspended in fresh media. The chondrocytes were cultured in high-density monolayer at 37 °C with 5% $CO_2$.

**Mice.** C57BL/6 male and NSG mice were obtained from Jackson Laboratory. NOD/MrkBomTac-Prkdcscid mice were obtained from Taconic Biosciences. Mice were housed and maintained in the Veterinary Medical Unit at the Veterans Affairs Palo Alto Health Care Systems. The Administrative Panel on Laboratory Animal Care of Stanford University approved animal protocols.

**Human skeletal muscle specimens.** The human muscle biopsy specimens were taken after patients (10–30 years, $n = 2$; 30–55 years, $n = 2$; 60–80 years, $n = 3$) gave informed consent as part of a human studies research protocol that was approved by the Stanford University Institutional Review Board. Sample processing for cell analysis began within 1–12 h of specimen isolation. In all studies, standard deviation reflects variability in data derived from studies using true biological replicates (i.e., unique donors). Data were not correlated with donor identity.

**MuSC isolation and purification.** Muscles were harvested from mouse hind limbs ($n = 4$) and mechanically dissociated to yield a fragmented muscle suspension. This was followed by a 45–50-min digestion in a Collagenase II-Ham's F10 solution (500 U ml$^{-1}$, Invitrogen). After washing, a second digestion was performed for 30 min with Collagenase II (100 U ml$^{-1}$) and Dispase (2 U ml$^{-1}$, ThermoFisher). The resulting cell suspension was washed, filtered, and stained with VCAM-biotin, CD31-FITC, CD45-APC, and Sca-1-Pacific-Blue antibodies, all at dilutions of 1:100. Human MuSCs were purified from fresh operative samples. Operative samples were carefully dissected from adipose and fibrotic tissue and a dissociated muscle suspension prepared as described for mouse tissue. The resulting cell suspension was then washed, filtered, and stained with anti-CD31-Alexa Fluor 488, anti-CD45-Alexa Fluor 488, anti-CD34-FITC, anti-CD29-APC, and anti-NCAM-Biotin antibodies. Unconjugated primary antibodies were then washed and the cells were incubated for 15 min at 4 °C in streptavidin-PE/Cy7 to detect NCAM-biotin. Cell sorting was performed on calibrated BD-FACSAria II or BD FACSAria III flow cytometers equipped with 488-, 633-, and 405-nm lasers to obtain the MuSC population. A small fraction of sorted cells was plated and stained for Pax7 and MyoD to assess the purity of the sorted population.

**mRNA transfection.** Cells were transfected using either mRNA-In (mTI Global Stem) for fibroblasts and chondrocytes, to reduce cell toxicity, or Lipofectamine MessengerMax (ThermoFisher) for endothelial cells and MuSCs, which were more difficult to transfect, using the manufacturer's protocol. For fibroblast and endothelial cells, serum free Pluriton medium with bFGF was used for transfection, while muscle stem cells and chondrocytes were kept in their original media—the former lacking serum and the later requiring serum to prevent the natural dedifferentiation of chondrocytes in culture. Culture medium was changed for fibroblasts and endothelial cells 4 h after transfection, but not for chondrocytes or MuSCs as overnight incubation was needed to produce a significant uptake of mRNA. Efficiency of delivery was confirmed by both GFP mRNA and immunostaining for individual factors in OSKMLN cocktail, the former also being used as a transfection control with the same protocol.

**Immunocytochemistry.** Cells were washed with HBSS/CA/MG and then fixed with 15% paraformaldehyde in PBS for 15 min. Cells were then blocked for 30 min to 1 h with a blocking solution of 1% BSA and 0.3% Triton X-100 in PBS for fibroblasts, endothelial cells, and 20% donkey serum/0.3% Triton in PBS for MuSCs. Primary antibodies were then applied in blocking solution and allowed to incubate overnight at 4 °C. The following day, the cells were washed with HBSS/CA/MG or PBST for MuSCs before switching to the corresponding Alexa Fluor-labeled secondary antibodies and incubated for 2 h. The cells were then washed again and stained with DAPI for 30 min and switched to HBSS/CA/MG for imaging or Fluoview for MuSCs.

**Autophagosome formation staining.** Cells were washed with HBSS/Ca/Mg and switched to a staining solution containing a proprietary fluorescent autophagosome marker (Sigma). The cells were then incubated at 37 °C in 5% $CO_2$ for 20 min, washed two times using HBSS/Ca/Mg, and stained for 15 min using CellTracker Deep Red cell labeling dye. Cells were then switched to HBSS/Ca/Mg for single-cell imaging using the Operetta High-Content Imaging System (Perkin Elmer).

**Proteasome activity measurement.** Wells were first stained with PrestoBlue Cell Viability dye (Life Technologies) for 10 min. Well signals were read using a TECAN fluorescent plate reader as a measure of cell count. Then cells were washed with HBSS/Ca/Mg before switching to original media containing the chymotrypsin-like fluorogenic substrate LLVY-R110 (Sigma), which is cleaved by

proteasome 20S core particle. Cells were then incubated at 37 °C in 5% $CO_2$ for 2 h before signals were again read on the TECAN fluorescent plate reader. Readings were then normalized by PrestoBlue cell count.

**Mitochondrial membrane potential staining.** Tetramethylrhodamine Methyl Ester Perchlorate (Thermo) was added to cell culture media. This dye is sequestered by active mitochondria based on their membrane potential. Cells were incubated for 30 min at 37 °C in 5% $CO_2$ and washed two times with HBSS/Ca/Mg before staining for 15 min using CellTracker Deep Red. Finally, cells were imaged in fresh HBSS/Ca/Mg using the Operetta High-Content Imaging System (Perkin Elmer).

**Mitochondrial ROS measurement.** Cells were washed with HBSS/Ca/Mg and then switched to HBSS/Ca/Mg containing MitoSOX (Thermo), a live-cell-permeant fluorogenic dye that selectively targeted to mitochondria and fluoresces when oxidized by superoxide. Cells were incubated for 10 min at 37 °C in 5% $CO_2$. Cells were then washed twice with HBSS/Ca/Mg, and stained for 15 min using CellTracker Deep Red. Finally, cells were imaged in fresh HBSS/Ca/Mg using the Operetta High-Content Imaging System (Perkin Elmer).

**SAβGal histochemistry.** Cells were washed twice with PBS then fixed with 15% Paraformaldehyde in PBS for 6 min. Cells were rinsed three times with PBS before staining with X-gal chromogenic substrate, which is cleaved by endogenous Beta galactosidase. Plates were kept in the staining solution, Parafilmed, to prevent from drying out, and incubated overnight at 37 °C with ambient $CO_2$. The next day, cells were washed again with PBS before switching to a 70% glycerol solution for imaging under a Leica bright-field microscope.

**Fixed and live-cell imaging.** Samples were imaged using fluorescent microscopes—the Operetta High-Content Imaging System (Perkin Elmer) or the BZ-X700 (Keyence)—and either a 10× or 20× air objective. Harmony (Operetta) or Volocity (BZ-X700) imaging software was used to adjust excitation and emission filters and came with preprogrammed Alexa Fluor filter settings which were used whenever possible. All exposure times were optimized during the first round of imaging, and then kept constant through all subsequent imaging.

**Image analysis.** Columbus (Operetta) or Image J (BZ-X700) was used for image analysis. Columbus software was to identify single cells utilizing DAPI of Cell-Tracker Re d to delineate nuclear and cell boundaries and calculate the signal statistics for each cell. Image J was used for muscle fibers to calculate the percentage of area composed of collagen by using the color threshold plug-in to create a mask of only the area positive for collagen. That area was then divided over the total area of the sample, which was found using the free draw tool. All other fiber analyses were performed using Volocity software and manually counting fibers using the free draw tool.

**Statistics.** Statistical analysis for physiological hallmarks of aging was done as described previously in Miller et al.[2]. Briefly, 100 cells were randomly selected from each experimental group (data depicted in Supplementary Figs. 2–5), and they were then pooled in a unique population of 800 cells for aged fibroblasts (100 cells × 8 individuals for both aged and aged treated); 300 cells for young fibroblasts (100 cells × 3 individuals); 700 cells for aged endothelial cells (100 cells × 7 individuals for both aged and aged treated); 300 cells for young endothelial cells (100 cells × 3 individuals). Box distribution plots display the fluorescence intensity quantification of 100 cells from each patient. Distributions were compared by statistical analysis by using multiple-comparison ANOVA. Arbitrary units for frequency distributions of different cell types should not be compared because staining was performed at different times. Matlab 2017 (MathWorks) was used for data presentation and analysis.

**Cytokine profiling.** This work was performed together with the Human Immune Monitoring Center at Stanford University. Cell media was harvested and spun at 400 rcf for 10 min at room temperature. The supernatant was then snap frozen with liquid nitrogen until analysis. Analysis was done using the human 63-plex kit (eBiosciences/Affymetrix). Beads were added to a 96-well plate and washed in a Biotek ELx405 washer. Samples were added to the plate containing the mixed antibody-linked beads and incubated at room temperature for 1 h followed by overnight incubation at 4 °C with shaking. Cold and room temperature incubation steps were performed on an orbital shaker at 500–600 rpm. Following the overnight incubation, plates were washed in a Biotek ELx405 washer and then biotinylated detection antibody added for 75 min at room temperature with shaking. Plates were washed as above and streptavidin-PE was added. After incubation for 30 min at room temperature, wash was performed as above and reading buffer was added to the wells. Each sample was measured in duplicate. Plates were read using a Luminex 200 instrument with a lower bound of 50 beads per sample per cytokine. Custom assay Control beads by Radix Biosolutions were added to all wells.

**Antibodies**. The following antibodies were used in this study. The source of each antibody is indicated. Rabbit::H3K9me3 (Abcam #ab8898 1:4000), LAP2α (Abcam #ab5162 1:500), SIRT1 (Abcam #ab7343 1:200); Rabbit: Mouse: HP1γ (Millipore Sigma #05-690 1:200), Lamin A/C (Abcam #ab40567 1:200), GFP (Invitrogen, #A11122, 1:250); Luciferase (Sigma-Aldrich, #L0159, 1:200); Collagen I (Cedarlane Labs, #CL50151AP, 1:200); HSP47 (Abcam, #ab77609, 1:200), Laminin (Abcam, #AB11576, 1:1000), anti-CD31-Alexa Fluor 488 (clone WM59; BioLegend; #303110, 1:75), anti-CD45-Alexa Fluor 488 (clone HI30; Invitrogen; #MHCD4520, 1:75), anti-CD34-FITC (clone 581; BioLegend; #343503, 1:75), anti-CD29-APC (clone TS2/16; BioLegend; #303008, 1:75) and anti-NCAM-Biotin (clone HCD56; BioLegend; #318319, 1:75), anti-CD31-Alexa Fluor 488 (clone WM59; BioLegend; #303110, 1:75), anti-CD45-Alexa Fluor 488 (clone HI30; Invitrogen; #MHCD4520, 1:75), anti-CD34-FITC (clone 581; BioLegend; #343503, 1:75), anti-CD29-APC (clone TS2/16; BioLegend; #303008, 1:75), and anti-NCAM-biotin (clone HCD56; BioLegend; #318319, 1:75).

**RNA sequencing and data analysis**. Cells were washed and digested by TRIzol (Thermo). Total RNA was isolated using the Total RNA Purification Kit (Norgen Biotek Corp) and RNA quality was assessed by the RNA analysis screentape (R6K screentape, Agilent); RNA with RIN > 9 was reverse transcribed to cDNA. cDNA libraries were prepared using 1 μg of total RNA using the TruSeq RNA Sample Preparation Kit v2 (Illumina). RNA quality was assessed by an Agilent Bioanalyzer 2100; RNA with RIN > 9 was reverse transcribed to cDNA. cDNA libraries were prepared using 500 ng of total RNA using the TruSeq RNA Sample Preparation Kit v2 (Illumina) with the added benefit of molecular indexing. Prior to any PCR amplification steps, all cDNA fragment ends were ligated at random to a pair of adapters containing a 8-bp unique molecular index. The molecular indexed cDNA libraries were than PCR amplified (15 cycles) and then QC'ed using a Bioanalzyer and Qubit. Upon successful QC, they were sequenced on an Illumina Nextseq platform to obtain 80-bp single-end reads. The reads were trimmed by 2 nt on each end to remove low-quality parts and improve mapping to the genome. The 78-nt reads that resulted were compressed by removing duplicates, while keeping track of how many times each sequence occurred in each sample in a database. The unique reads were then mapped to the human genome using exact matches. This misses reads that cross exon–exon boundaries, as well as reads with errors and SNPs/ mutations, but it does not have substantial impact on estimating the levels of expression of each gene. Each mapped read was then assigned annotations from the underlying genome. In case of multiple annotations (e.g., a miRNA occurring in the intron of a gene), a hierarchy based on heuristics was used to give a unique identity to each read. This was then used to identify the reads belonging to each transcript and coverage over each position on the transcript was established. This coverage is nonuniform and spiky. Therefore, we used the median of this coverage as an estimate of the expression value of each gene. In order to compare the expression levels in different samples, quantile normalization was used. Further data analysis was done in Matlab. Ratios of expression levels were then calculated to estimate the log (base 2) of the fold change. Student's $t$ test was used to determine significance with a $p < 0.05$ cutoff. Molecular Signatures Database categorization was done using Broad Institute online tools https://software.broadinstitute.org/gsea/msigdb/.

**Gene expression analysis**. Total RNA was purified using the RNeasy Plus Mini kit (Qiagen), and cDNA was prepared with the First-strand cDNA synthesis kit (Applied Biosystems). The quantitative polymerase chain reaction was performed using VeriQuest Mastermix (ThermoFisher Scientific) for SYBR Green and Taq-man primer sets, respectively. The relative gene expression was analyzed by the ΔΔCt method and normalized to glyceraldehyde-3-phosphate dehydrogenase (GAPDH). The Taqman probes for human GAPDH (Hs02758991); COL2A1 (Hs00264051); SOX9 (Hs00165814); MMP3 (Hs00233962) and MMP13 (Hs00233992) were purchased from Applied Biosystems. The SYBR green primer sequences used are: Human SOD2 (F)-5′GGC CTA CGT GAA CAA CCT GA3′; Human SOD2 (R)-5′TGG GCT GTA ACA TCT CCC TTG3′; Human iNOS (F)-5′ GTC CCG AAG TTC TCA GGC CA3′; Human iNOS (R)-5′GTT CTT CAC TGT GGG GCT TG3′; Human RANKL (F)-5′CAG GTT GTC TGC AGC GT3′ and Human RANKL (R)-5′GAT CCA TCT GCG CTC TGA AAT A3′; Human GAPDH (F)- 5′TGT CCC CAC TGC CAA CGT GTC3′; Human GAPDH (R)-5′ AGC GTC AAA GGT GGA GGA GTG GGT3′.

**ATP assay**. ATP in the chondrocytes was measured using colorimetric assay and the ATP assay kit (ab83355; Abcam, Cambridge, MA) following the manufacturer's instructions. Cells were washed in cold phosphate buffered saline and homogenized and centrifuged to collect the supernatant. The samples were loaded with assay buffer in triplicate. ATP reaction mix and background control (50 μL) was added to the wells and incubated for 30 min in dark. The plate was read at OD 570 nm using SpectraMax M2e (Molecular Devices, Sunnyvale, CA). The mean optical density was used to estimate of the intracellular ATP concentration relative to the standard curve.

**Cell proliferation assay**. Cell viability was assayed using the PrestoBlue Cell Viability (Life Technologies) reagent consecutively for 3 days post transfection in accordance with the manufacturer's instructions. PrestoBlue reagent was added to the cell culture medium, and the cells were incubated at 30 °C for 30 min. Absorbance of the PrestoBlue was measured daily using SpectraMax M2e (Molecular Devices, Sunnyvale, CA).

**EDU staining**. Staining was done according to the manufacturer's protocol using the Click-iT EdU kit. Cells were labeled with Edu after switching to growth media. Cells were allowed to grow 1 or 2 days before fixation with 4% paraformaldehyde and permeabilization with 0.5% Triton X-100 in PBST. Cells were the incubated in Click-It reaction cocktail for 30 min before washing in PBS and imaging.

**MitoTracker staining and flow cytometry analysis**. Cells were washed twice with Ham's F10 (no serum or pen/strep). Subsequently, MuSCs were stained with MitoTracker Green FM (ThermoFisher, M7514) and DAPI for 30 min at 37 °C, washed three times with Ham's F10, and analyzed using a BD FACSAria III flow cytometer.

**Myogenic colony-forming cell assay for MuSCs**. Single treated and control MuSCs were deposited into wells of collagen- and laminin-coated plates at one cell per well by BD FACSAria III flow cytometer. Collagen/laminin coating was accomplished by overnight incubation of the plates rocking at 4 °C with a 1:1 mixture of laminin (10 μg/ml ThermoFisher 23017-015) and collagen (10 μg/ml Sigma C8919) in PBS. Coated wells were washed three times with PBS before use. The cells were cultured in grow media, F10 medium supplemented with 20% horse serum, and 5 ng/ml basic fibroblast growth factor (bFGF; PeproTech 100-18B). After 6 days of culture, plates were fixed with 4% paraformaldehyde (Electron Microscopy Services 15710), stained with DAPI (Invitrogen D1306), and scored by microscopy to determine the number of myogenic colony-forming cells, defined by wells that contained at least eight cells.

**Myogenic/fusion index**. Myogenic analysis was completed as previously described[36]. After MuSCs underwent reprogramming or control treatment, cells were cultured in grow media. To induce differentiation, myoblast cultures were maintained in DMEM supplemented with 2% horse serum. The myogenic/fusion index was determined as the percentage of myonuclei in myotubes (defined as cells with three or more nuclei) compared with the total number of nuclei in the field.

**Lentiviral transduction**. Luciferase and GFP protein reporters were subcloned into a third-generation HIV-1 lentiviral vector (CD51X DPS, SystemBio). To transduce freshly isolated MuSCs, cells were plated on a Poly-D-Lysine (Millipore Sigma, A-003-E) and ECM coated eight-well chamber slide (Millipore Sigma, PEZGS0896) and were incubated with 5 μl of concentrated virus per well and 8 μg/mL polybrene (Santa Cruz Biotechnology, sc-134220). Plates were spun for 5 min at 3200 g, and for 1 h at 2500 g at 25 °C. Cells were then washed with fresh media two times, scraped from plates, and resuspended in the final volume according to the experimental conditions.

**Bioluminescence imaging**. Bioluminescent imaging was performed using the Xenogen IVIS-Spectrum System (Caliper Life Sciences). Mice were anesthetized using 2% isoflurane at a flow rate of 2.5 l/min Intraperitoneal injection of D-Luciferin (50 mg/ml, Biosynth International Inc.) dissolved in sterile PBS was administered. Immediately following the injection, mice were imaged for 30 s at maximum sensitivity (f-stop 1) at the highest resolution (small binning). Every minute, a 30-s exposure was taken until the peak intensity of the bioluminescent signal began to diminish. Each image was saved for subsequent analysis.

**Bioluminescence image analysis**. Analysis of each image was performed using Living Image Software, version 4.0 (Caliper Life Sciences). A manually generated circle was placed on top of the region of interest and resized to completely surround the limb or the specified region on the recipient mouse. Similarly, a background region of interest was placed on a region of a mouse outside the transplanted leg.

**Tissue harvesting**. TiA muscles were carefully dissected away from the bone, weighed, and placed into a 0.5% PFA solution for fixation overnight. The muscles were then moved to a 20% sucrose solution for 3 h or until muscles reached their saturation point and began to sink. The tissues were then embedded and frozen in Optimal Cutting Temperature (OCT) medium and stored at −80 °C until sectioning. Sectioning was performed on a Leica CM3050S cryostat that was set to generate 10-μm sections. Sections were mounted on Fisherbrand Colorfrost slides. These slides were stored at −20 °C until immunohistochemistry could be performed.

**Flow cytometry**. For mouse MuSC sorting scheme, we followed the same gating strategy previously published[37]. For human MuSC sorting scheme, we followed the same strategy previously published[38].

**Histology**. TiA muscles were fixed for 5 h using 0.5% electron-microscopy-grade paraformaldehyde and subsequently transferred to 20% sucrose overnight. Muscles were then frozen in OCT, cryosectioned at a thickness of 10 μm, and stained. For colorimetric staining with Hematoxylin and Eosin (Sigma) or Gomori Trichrome (Richard-Allan Scientific), samples were processed according to the manufacturer's recommended protocols.

**Ex vivo force measurement**. To measure the force, we isolated the TiA in a bath of oxygenated Ringer's solution and stimulated it with plate electrodes. Immediately after euthanasia, the distal tendon of the TiA, the TiA, and the knee (proximal tibia, distal femur, patella, and associated soft tissues) were dissected out and placed in Ringer's solution (Sigma) maintained at 25 °C with bubbling oxygen with 5% carbon dioxide. The proximal tibia was sutured to a rigid wire attached to the force transducer, and the distal tendon was sutured to a rigid fixture. No suture loops or slack was present in the system. The contralateral limb was immediately dissected and kept under low passive tension in oxygenated Ringer's solution bath until measurement. Supramaximal stimulation voltage was found, and the active force-length curve was measured in a manner similar to the in vivo condition. After measurement, the muscle was dissected free and the mass measured. An Aurora Scientific 1300-A Whole Mouse Test System was used to gather force production data.

**DNA methylation data**. The human Illumina Infinium EPIC 850K chip was applied to $n = 16$ DNA samples (corresponding to two treatment levels (before/after treatment) of four fibroblasts and four endothelial cells). The raw image data were normalized using the "preprocessQuantile" normalization method implemented in the "minfi" R package[39,40].

**Epigenetic clock analysis**. Several DNAm-based biomarkers have been proposed in the literature, which differ in terms of their applicability (most were developed from blood), and in terms of their biological interpretation (reviewed in ref. [11]). We focused on two epigenetic clocks that apply to fibroblasts and endothelial cells. In our primarily analysis, we used the pan-tissue epigenetic clock[3] because it applies to all sources of DNA (with the exception of sperm). A previously defined mathematical algorithm is used to combine the methylation levels of 353 CpG into an age estimate (in units of years), which is referred to as epigenetic age or DNAm age[3]. In our secondary analysis, we used the skin-and-blood epigenetic clock (based on 391 CpGs) because it is known to lead to more accurate DNAm age estimates in fibroblasts, keratinocytes, buccal cells, blood cells, saliva, and endothelial cells[13].

We used the online version of the epigenetic clock software to arrive at DNA methylation age estimates from $n = 16$ samples collected from $n = 8$ individuals[3]. Although the chronological age range was relatively narrow (ranging from 47 to 69 years, median age = 55), the two DNAm age estimates exhibited moderately high correlations with chronological age ($r = 0.42$ and $r = 0.63$, $P = 0.0089$ for the pan-tissue- and the skin-and-blood clock, respectively).

Two samples (before and after rejuvenation treatment) were generated from each of $n = 8$ individuals. To properly account for the dependence structure in the data, we used linear mixed effects models to regress DNAm age (dependent variable) on treatment status, chronological age, and individual identifier (coded as random effect). Toward this end, we used the "lmer" function in the "lmerTest" R package[41].

**Reporting summary**. Further information on research design is available in the Nature Research Reporting Summary linked to this article.

## Data availability

The data that support the findings of this study are available from the corresponding author upon request. The data used for the methylation clock analysis will be available through the following GSE number starting April 08 2020: GSE142439. RNASeq data have been deposited to the Sequence read Archive (SRA), and will be available upon publication through the following SRA number: PRJNA598923.

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

## Acknowledgements

We thank the members of the Sebastiano, Bhutani, and Rando laboratories for comments and discussions; we thank Jens Durruthy for valuable technical information on mRNA-based transient reprogramming. We thank Tony Wyss-Coray and his lab for providing primary fibroblast lines from patients diagnosed with AD, and Helen Blau and her lab for technical consulting on telomere-length evaluation. This work was supported by the AFAR Junior Investigator Award to V.S., by grants from the National Institutes of Health (NIH/NIAMS) (R01 AR070865 and R01 AR070864) to N.B., the Glenn Foundation for Medical Research and by grants from the National Institutes of Health (NIH) (P01 AG036695, R01 AG23806 (R37 MERIT Award), R01 AG057433, and R01 AG047820), and the Department of Veterans Affairs (BLR&D and RR&D Merit Reviews) to T.A.R.; and by the CalPoly funding Award # TB1-01175.

## Author contributions

V.S., T.A.R., T.J.S., N.B., and M.Q. conceived, designed, and supervised the experiments reported. T.J.S. performed the Fibroblast and Endothelial in vitro experiments. T.J.S. performed the processing for the Fibroblast and Endothelial RNA sequencing experiments. N.B. and C.C. obtained the cartilage surgical specimens and S.M and T.J.S performed the chondrocyte isolation and in vitro experiments. P.P., C.M.T., and A.C. performed the MuSC isolation, lentivirus infection design, and experiments. T.J.S., P.P., C.M.T., and A.C. performed the MuSCs in vitro experiments. M.Q., P.P., A.C., and L.D. performed the MuSCs in vivo experiments. T.J.S., S.M., N.B., M.Q., C.C., S.H., L.S.Q., T.A.R., and V.S. analyzed data and wrote the paper.

## Competing interests

The authors declare no competing interests.
