## [Peer Review File · Nature Communications]

Editorial Note: This manuscript has been previously reviewed at another journal that is not operating a transparent peer review scheme. This document only contains reviewer comments and rebuttal letters for versions considered at Nature Communications .

REVIEWERS' COMMENTS:

Reviewer #1 (Remarks to the Author):

Authors have satisfactorily addressed my concerns. I think that this is a valuable step forward on a topic that I suspect we are very far from understanding.

1. In the current version, I have noticed a few changes in the data relative to the previous version. I imagine that this is due to the increased number of samples, is it?. For example, cytokines levels in endothelial cells have changed remarkably between former Fig. 1i and current Fig. 2j (cytokine IL18 is notable; was not significantly changed before and it is dramatically changed now; GMCSF, that was significant before is not shown now; etc).
2. Despite the indicated limitations of the Ocampo paper, I would not say that the evidence they presented is "preliminary" (line 63).
3. The Ocampo paper is misrepresented in the Introduction when it says that they do not demonstrate "whether the rejuvenation is a direct and cell intrinsic effect". This is not correct, the Ocampo paper has three figures (1, 2 and 6) devoted to in vitro rejuvenation of cultured mouse and human cells, very much like in the current paper. Please, acknowledge this in the Introduction.

Reviewer #2 (Remarks to the Author):

The authors have answered some of the technical points previously raised; what is left is the important issue as to whether the effects shown are actually reverting aging mechanisms, or are simply reprogramming cells to iPS cells, thus re-starting the aging clock. The latter might still be relevant for actual treatment, but at some risk, and not really that different than what Ocampo et al have already done.

As for the sarcopenia data, it is still very surprising that one could revert age-related strength decrements in such a way, given the prior data from several labs as to niche-effects on satellite cells, due to senescent cells, among other factors. But it doesn't seem fair to these authors to simply say one doesn't believe the data; it will be up to others to see if these findings are reproducible.

Reviewer #4 (Remarks to the Author):

I think that the authors have done an good amount of work to answer the reviewer 3's questions (as well as the other raised questions and concerns). I agree with the reviewer that the manuscript lacks mechanism, however I think that this can be solved in the future. Contrary to Reviewer 3's opinion, I believe that it is a strength the fact that they look at several cell types. Indeed, I think that the observations have been made broad enough to be of general interest and are valid and sufficient for publication in Nature Communications.

In regard to the applied statistics, this is a difficult aspect of the paper. I am not sure if there is a consensus way to analyse the data due to the multiple variables and sampling in the paper. Nevertheless, I think that with the new analysis and Supplementary Figures, the authors provide

enough transparency on regards to their findings, therefore readers can make their own analysis and conclusions. On this note, due to this complexity, I really advice that the authors make all their data accessible to the public (including the processed data, such as tables with the data used for their plots. Indeed, this is mandatory in Nature family journals).

Although I do not believe that the authors addressed correctly the question regarding senescence/proliferation, please include the EdU incorporation assay, since it will provide some light into this aspect.

Reviewer #1 (Remarks to the Author):

Authors have satisfactorily addressed my concerns. I think that this is a valuable step forward on a topic that I suspect we are very far from understanding.

We want to thank the reviewer for the points he/she raised in the first place. We are glad we have addressed all the concerns and we agree that this topic is far from understood. We are currently heavily working on follow up experiments to better understand the mechanism of this process.

1. In the current version, I have noticed a few changes in the data relative to the previous version. I imagine that this is due to the increased number of samples, is it?. For example, cytokines levels in endothelial cells have changed remarkably between former Fig. 1i and current Fig. 2j (cytokine IL18 is notable; was not significantly changed before and it is dramatically changed now; GM-CSF, that was significant before is not shown now; etc).

The changes in significance are due, as correctly pointed out, by the increased n size.

2. Despite the indicated limitations of the Ocampo paper, I would not say that the evidence they presented is “preliminary” (line 63).

We agree and we have replaced the word “preliminary” with the word “first” in the manuscript.

3. The Ocampo paper is misrepresented in the Introduction when it says that they do not demonstrate “whether the rejuvenation is a direct and cell intrinsic effect”. This is not correct, the Ocampo paper has three figures (1, 2 and 6) devoted to in vitro rejuvenation of cultured mouse and human cells, very much like in the current paper. Please, acknowledge this in the Introduction.

We would like to highlight that in Ocampo et al the human cells were in vitro differentiated from hiPSCs and by all means such cells cannot be considered a model of in vivo aging since in vitro aging is less complex and cells in vitro differentiated from iPSCs are considered fetal-like. Conversely, we have used primary cultures from naturally aged individuals. Nonetheless, we have addressed this in the introduction and deleted the sentence “The authors used mice where activation of the reprogramming factors was systemic, without demonstrating whether the rejuvenation is a direct and cell intrinsic effect or rather the result of extrinsic rejuvenating factors that are expressed and secreted by some cells in response to OSKM upregulation and that promote

rejuvenation systemically”

Reviewer #2 (Remarks to the Author):

The authors have answered some of the technical points previously raised; what is left is the important issue as to whether the effects shown are actually reverting aging mechanisms, or are simply reprogramming cells to iPS cells, thus re-starting the aging clock. The latter might still be relevant for actual treatment, but at some risk, and not really that different than what Ocampo et al have already done.

We thank the reviewer for the positive feedback. Our work is a significant step forward to what has been shown by Ocampo et al. Our work is almost entirely focused on naturally aged human cells. We provide the first comprehensive analysis of the process of rejuvenation across different cell types and different hallmarks of aging.

As for the sarcopenia data, it is still very surprising that one could revert age-related strength decrements in such a way, given the prior data from several labs as to niche-effects on satellite cells, due to senescent cells, among other factors. But it doesn't seem fair to these authors to simply say one doesn't believe the data; it will be up to others to see if these findings are reproducible.

We agree. The results are absolutely surprising. Our approach is different from others', so the direct comparison is difficult to do. Our results consistently indicate that if ex-vivo rejuvenated by transient and short expression of OSKMNL and then transplanted into an injured muscle, satellite cells can regenerate the muscle as young cells and restore tetanic force of young uninjured muscle.

Reviewer #4 (Remarks to the Author):

I think that the authors have done an good amount of work to answer the reviewer 3's questions (as well as the other raised questions and concerns). I agree with the reviewer that the manuscript lacks mechanism, however I think that this can be solved in the future. Contrary to Reviewer 3's opinion, I believe that it is a strength the fact that they look at several cell types. Indeed, I think that the observations have been made broad enough to be of general interest and are valid and sufficient for publication in Nature Communications.

We want to thank the reviewer for the positive assessment of our work.

In regard to the applied statistics, this is a difficult aspect of the paper. I am not sure if there is a consensus way to analyse the data due to the multiple variables and sampling in the paper. Nevertheless, I think that with the new analysis and Supplementary

Figures, the authors provide enough transparency on regards to their findings, therefore readers can make their own analysis and conclusions. On this note, due to this complexity, I really advice that the authors make all their data accessible to the public (including the processed data, such as tables with the data used for their plots. Indeed, this is mandatory in Nature family journals).

We agree and will provide all the raw data used to compile all the Figures upon request. We have deposited RNASeq data and methylation data on repositories; links to data source will be available to the readers.

Although I do not believe that the authors addressed correctly the question regarding senescence/proliferation, please include the EdU incorporation assay, since it will provide some light into this aspect.

We have included this plot in supplementary information (Supplementary Figure 8)